# What happened to health labour markets during COVID-19? Insights from the analysis of cross-sectional survey data on the perceptions of medical doctors in Brazil

Giuliano Russo [1], Bruno Luciano Carneiro Alves Luciano de Oliveira [2], Alex J Flores Cassenote,[3] Mário C Scheffer [4]

[1]Wolfson Institute of Population Health, Queen Mary University of London, London, UK
[2]Curso de Medicina, Universidade Federal do Maranhao, Sao Luis, Maranhão, Brazil
[3]Departamento da Medicina Preventiva, Universidade de São Paulo Hospital das Clínicas, Sao Paulo, Brazil
[4]Preventative Medicine, Universidade de Sao Paulo Faculdade de Medicina, Sao Paulo, Brazil

**Correspondence to**
Dr Giuliano Russo;
g.russo@qmul.ac.uk

## ABSTRACT

**Objectives** To examine physicians' perceptions of changing employment opportunities in Brazil, and gain an insight into labour markets in low/middle-income countries (LMICs) during the pandemic.
**Study design** Descriptive and inferential analysis of a quantitative dataset from a representative cross-sectional survey of physicians of two Brazilian states.
**Settings** São Paulo and Maranhão states in Brazil.
**Participants** Representative sample of 1183 physicians.
**Outcome measures** We estimated prevalence and 95% CIs for physicians' perceptions of changes in demand and supply of doctors, as well as changes in prices of medical services for facilities of practice in the two states, stratified by public, private and dual-practice physicians.
**Results** Most doctors reported increased job opportunities in the public sector (54.9%, 95% CI 52.0% to 57.7%), particularly in Maranhão state (65.0%, 95% CI 60.9% to 68.9%). For the private sector, increased opportunities were reported only in large private hospitals (46.7%, 95% CI 43.9% to 49.6%) but not in smaller clinics. We recorded perceptions of slight increases in availability of doctors in Maranhão, particularly in the public sector (51.4%, 95% CI 43.2% to 59.5%). Younger doctors recounted increased vacancies in the public sector (64%, 95% CI 58.1% to 68.1%), older doctors only in walk-in clinics in Maranhão (47.5%, 95% CI 39.9% to 55.1%). Those working directly with patients with COVID-19 saw opportunities in public hospitals (65%, 95% CI 62.3% to 68.4%) and in large private ones (55%, 95% CI 51.8% to 59.1%).
**Conclusions** Our findings hint that health labour markets in LMICs may not necessarily shrink during epidemics, and that impacts will depend on the balance of public and private services in national health systems.

## INTRODUCTION AND BACKGROUND

The COVID-19 pandemic and the ensuing economic crisis have had an unparalleled impact on population health,[1] national economies[2] and working modalities.[3] A recent Organisation for Economic Co-operation and Development report presents evidence on the ways labour markets have changed due to pandemic-related lockdown measures, as businesses were disrupted and workers furloughed or laid off.[4]

The economic literature has so far suggested that health labour markets are 'recession proof', that is, one of the few sectors to hold up during economic slowdowns.[5] Scholars have argued this may be due to an inelastic demand for healthcare services, and to increasing health needs during recessions.[6] Evidence from the USA in fact showed increasing employment opportunities in the healthcare industry during the 2008 financial crisis, particularly for nurses.[7] Studies from Canada and Australia concluded that labour markets for physicians were largely unaffected during the ensuing recession.[8 9] However, in those countries with publicly funded health systems, government-driven austerity measures did reduce health sector resources and shrink health labour markets, particularly in Europe.[10–12]

**STRENGTHS AND LIMITATIONS OF THIS STUDY**
⇒ This paper is part of a wider research project on the impact of COVID-19 on physicians.
⇒ We employ a dataset from a representative sample of physicians from two Brazilian states.
⇒ We used labour market theory to interpret changes in availability of doctors during epidemics.
⇒ However, our analysis is based on physicians' perceptions, which may not be objective measures of changes in labour markets.
⇒ São Paulo and Maranhão states may not be entirely representative of wider Brazil or other low/middle-income countries.

There is a growing body of literature highlighting the role of labour markets in the provision of healthcare services in both high-income countries and low/middle-income countries (LMICs),[13 14] which tends to advocate for the use of health labour market analysis to design effective health policies in such contexts.[15 16]

During COVID-19, research has shown that global labour health markets experienced a multilayered crisis amplified by the combination of lockdown measures and reduced worker mobility.[17] In the first year of the pandemic, studies from the USA argued that health sector unemployment rose because of pay cuts and redundancies, as patients were delaying seeking treatment and hospitals were focusing on COVID-19 care, which 'is not where the money is'.[18] Other US scholars[19] noticed that unemployment in the healthcare industry increased less than in other sectors during COVID-19; while less specialised jobs (such as non-healthcare hospital workers and therapists) were badly affected, employment opportunities for physicians barely shifted. As many patients were moved to government-funded schemes, new 'price sensitivities' were found in the demand for healthcare service, as customers would no longer be insulated from costs.[20]

The evidence from outside the USA is less consolidated. Analysis of labour statistics and employment censuses from the UK market[21] shows that healthcare employment declined suddenly in 2020 only to bounce back the year after, with dentists and nurses among the worst affected professions. Another British study looking at job ads during the pandemic[22] found more care work and nursing vacancies than in any other sector during lockdowns. Similar findings were reported for Serbia, where COVID-19 seems to have entrenched a continuous mismatch between supply and demand for physicians and nurses.[23] An online survey study from Iran[24] suggested that healthcare employment would have become less attractive because of COVID-19, as more health workers consider leaving. At the height of the first wave of the pandemic, there were reports[25] that Mexico, South Africa and Zambia were recruiting doctors from abroad, as a surge in the demand for COVID-19 care was expected.

Brazil has been one of the world's most affected countries, with over 700 000 COVID-19-related deaths; its economy contracted by 3.9% in 2020, although it rebounded by 4.6% and 2.6% in the following years.[26] The country's unemployment rate is currently at 9%, and the International Monetary Fund estimates that approximately 12 million jobs have been lost as a consequence of the pandemic, with a disproportionate impact among the lower-income groups.[27]

The impact on labour markets in each country depended essentially on the epidemiological spread of the disease, by the lockdown measures implemented, and by the structure and stage of development of the economy.[4] Past economic recessions historically had little repercussions for physician employment, as in Brazil demand has always outstripped supply.[28]

Lockdown measures and restrictions were introduced patchily and by the previous government in Brazil,[29] with a view not to disrupt the formal and informal economy. Initial shortcomings in the availability of tests hampered the collection of epidemiological data for SARS-CoV-2 infections among health professionals, and for the first year of the pandemic, there was no official testing protocol for the wider population or health workers.[30] Amid such an environment, elective medical procedures were suspended, but the vast majority of essential healthcare services kept functioning.[31] A recent study showed that public sector physicians' workload and earnings increased in Brazil during the first 2 years of the pandemic, while those of private doctors suffered.[32]

Brazil's healthcare system comprises a publicly funded Unified Health System (Sistema Único de Saúde, SUS) and a multiplicity of privately financed subsectors, including large, comprehensive private hospitals and smaller, outpatient care surgeries.[33] In the last 20 years, low-cost walk-in clinics in urban areas (Clínicas Populares or People's Clinics) have started to provide out-of-pocket private services, mostly outpatient in nature.[34 35] Within SUS, provision of free-at-the-point-of-care services is often outsourced to private entities (social health organisations) that manage public hospitals and contract their own staff, including doctors. Private funds account for more than half (54%) of Brazil's health spending, including out-of-pocket medicines and private insurance premiums.[36] Access to such private services is funded through employment-related health insurance plans, with 24.2% of Brazil's population owning such private plans.[37]

Previous analysis of this very dataset showed that a substantial proportion of physicians in our sample were infected with SARS-CoV-2 during the first year of the pandemic, that the infection rate in Maranhão was almost twice as that in São Paulo, and as being a physician in Maranhão, younger and having worked in a COVID-19 ward were characteristics associated with the probability of infection.[38] We also found that workloads and earnings of public doctors increased the most in the first year of the epidemic, particularly in Maranhão. Although earnings remained broadly stable in the public sector, one-third of public sector-only physicians in Maranhão (MA) saw an increase in their earnings, while more than half of private ones saw a decrease in their earnings.[32] As for working modalities, in the same sample, we also saw that telemedicine was employed by one-third of doctors as a modality to provide healthcare services, particularly for hospital-based inpatient services, more frequently in private clinics in São Paulo than in Maranhão.[39]

In this paper, we use physicians' perceptions of changing employment opportunities in Brazil to gain an insight into health labour markets during COVID-19 in LMICs. The aim of this study is to contribute to the existing body of work on health labour markets in mixed health systems across the world; this would provide an evidence base for policies to mitigate the effects of future shocks on health workforce.

## METHODS

### Methodological approach

As part of a wider study on the health workforce in two Brazilian states,[40] we conducted secondary analysis of a dataset from a representative cross-sectional survey on physicians' perceptions of the impact of COVID-19 on their health, earnings and work routines.[32 38 39] Workers' perceptions have been used before in the economic literature to explore labour market dynamics in high-income contexts.[41]

In our survey, doctors were asked whether they had observed in the past 2 years: (a) an increase or a decline of job opportunities and vacancies in their workplace; (b) any change in the availability of doctors; and (c) any change in the remuneration of a 12-hour shift in their institution's Accidents and Emergency (A&E) ward (see the survey questionnaire in online supplemental information 1). We interpreted the reported changes in (a) as proxies for changes in demand for doctors; responses on (b) as proxies for supply and responses on (c) as a proxy for changes in the level of prices for healthcare services.

As our doctors worked in either public or private facilities, or both, we considered the doctors' perceptions of changes in their own sector of employment as particularly accurate. As a way of validating responses, we triangulated the perceptions of the entire sample with those from doctors working specifically in such sectors. We outline the limitations of such an approach in the Discussion section.

### Study settings and data collection

The original survey was carried out in one rich state (São Paulo) and one less developed one (Maranhão) in Brazil, with a view to capture the differential effects of economic recessions in diverse health markets.[37] São Paulo state is home to more than 46.6 million people, has one of the country's highest per capita incomes and 38% of its population is covered by private health schemes. The public health expenditure is also among the highest, estimated at US$360.28 per capita in 2018 (US$650 Prices at Purchasing Parity (PPPr), and its medical workforce includes 163 430 physicians (3.5 per 1000 inhabitants, the third highest in the country). By contrast, Maranhão state is home to approximately 7.2 million people, its per capita income is one-third of São Paulo's and only 1% of the population is covered by private health schemes. In 2022, there were 8743 physicians in Maranhão, that is, 1.22 per 1000—the second lowest rate among Brazilian states.[42]

Our survey was conducted between 16 February and 15 June 2021. The sample was drawn from the nominal listing of physicians registered with Brazil's Federal Council of Medicine in the two states. The study's overall sample was composed of 1183 physicians, consisting of 632 from São Paulo and 551 from Maranhão. The sample was calculated based on the active physicians registered with the Federal Council of Medicine in the two states in 2021 (N=152 511—144 852 in São Paulo and 7659 in Maranhão) and their key demographic characteristics. Proportional stratified sampling was used to replicate the physician distribution by gender, age, state and residence. A larger proportion of Maranhão's pool of physicians was selected to allow for a sufficient N for the strata and doctor characteristics of interest. As the two states are very different—in São Paulo, there are 30% of all the physicians and in Maranhão, a little more than 1%—a proportional sample would have been too small to allow stratifications in Maranhão (see online supplemental table 1).

The survey was carried out by a specialised research institute (Datafolha Research Institute), under the technical supervision of the academic researchers. Primary data were collected via a telephone survey conducted in Portuguese by Datafolha data collectors, which included a field coordinator, experienced interviewers and administrative staff responsible for checking missing data. Sample size calculations, sample selection, questionnaire design, substitution control, database assembly and data analysis were performed by the authors of this paper.

### Patient and public involvement

Medical doctors as well as members of the public were consulted and participated in the design of the original version of the survey questionnaire. The questionnaire was then piloted by Datafolha in a subset of 10 doctors in the two states, and a final version was elaborated following the feedback received.

### Variables and data analysis

The interviews consisted of a 30-minute telephone questionnaire, containing 30 questions ranging from multiple, closed questions to interdependently concatenated and semiopen questions. The specific variables for this secondary analysis were constructed from the questions below from the Survey Questionnaire's Section III– Changes in the Labour Market (see survey questionnaire in online supplemental annex 1).

For our analysis in this study, the prevalence and 95% CIs of variables related to physicians' perceptions of changes in job opportunities and availability of doctors were estimated for the two states in their facilities of practice (public hospitals-SUS, private doctor surgeries, large private hospitals and walk-in clinics), and stratified for public-only physicians, private-only ones and dual practitioners (table 1).

P values were calculated based on $X^2$ tests. Statistically significant differences at the 5% confidence level were considered in the absence of overlapping 95% CIs. Prevalence and 95% CIs for such perceptions were also analysed and plotted by physicians' age groups (<35 years, 35–50 years, ≥50 years) and by their specific involvement with COVID-19 services. The database developed in Excel by the Datafolha data collectors was exported to R-Studio V.4.1.3 for statistical treatment.[43]

**Table 1** Variables, survey questions and categories used in the analysis

| Variable | Survey question | Categories |
|---|---|---|
| Demand for physicians in the public sector (SUS) | Q.24 In your opinion, comparing with before the pandemic, at SUS____ | 1. There are fewer work opportunities<br>2. There have been no significant changes in work opportunities OR<br>3. There are more work opportunities<br>4. Does not know |
| Demand for physicians in private doctor surgeries | Q.26 In your opinion, comparing with before the start of the pandemic, in private doctor surgeries ____ | 1. There are fewer work opportunities<br>2. There have been no significant changes in work opportunities OR<br>3. There are more work opportunities<br>4. Does not know |
| Demand for physicians in large private hospitals | Q.28 In your opinion, comparing with before the start of the pandemic, in large private hospitals ____ | 1. There are fewer work opportunities<br>2. There have been no significant changes in work opportunities OR<br>3. There are more work opportunities<br>4. Does not know |
| Demand for physicians in private walk-in clinics | Q.30 In your opinion, comparing with before the start of the pandemic, in private walk-in clinics ____ | 1. There are fewer work opportunities<br>2. There have been no significant changes in work opportunities OR<br>3. There are more work opportunities<br>4. Does not know |
| Supply of physicians | Q.32 In your opinion, comparing with before the pandemic, the availability of new physicians to fill vacancies ______ | 1. Has decreased<br>2. There have been no significant changes in the number of medical professionals available OR<br>3. Has increased<br>4. Does not know |
| Prices of healthcare services | Q.34 In your opinion, comparing with before the pandemic, the amount paid to physicians for a 12-hour A&E shift ______ | 1. Has decreased<br>2. There have been no significant changes in the amount paid to physicians in your specialty OR<br>3. Has increased<br>4. Does not know |

Source: USP-UFMA-QMUL (2022).
A&E, Accidents and Emergency; SUS, Sistema Único de Saúde; USP-UFMA-QMUL, University of São Paulo-Federal University of Maranhão-Queen Mary University of London.

## RESULTS

Our sample included 1183 physicians, 551 from Maranhão and 632 from São Paulo, located in urban as well as rural areas (see the graphical map of physicians' locations in online supplemental figure 1).

Most physicians in our survey (58.3%) worked concomitantly in public and private healthcare services, with only 19.6% of them employed exclusively in the public ones (see online supplemental table 1). Most of our physicians were deployed directly to the delivery of COVID-19 services, to COVID-19 wards or to COVID-19-specific outpatient care (63.4%). No significant difference was found in physicians' characteristics or employment across the two states.

Most doctors in our sample said job opportunities and vacancies in the public sector (SUS) increased during the COVID-19 pandemic (54.9%, 95% CI 52.0% to 57.7%). This was particularly evident among Maranhão doctors (65.0%, 95% CI 60.9% to 68.9%). Opportunities and vacancies were also said to have increased in large private hospitals with inpatient care capacity (46.7%, 95% CI 43.9% to 49.6%), although not as much as in SUS. Again, such positive perceptions were found to be more pronounced among Maranhão doctors (see table 2).

For other private sector facilities, perceptions of changing employment opportunities were either negative or neutral, particularly for smaller private doctor practices, where 57.2% of doctors declared opportunities to have actually decreased (95% CI 54.5 to 60.0). Such negative perceptions for smaller private clinics were especially acute in São Paulo (60.9%, 95% CI 57.1% to 64.7%) (table 2).

These perceptions were confirmed when separating the views of public and private sector doctors, as public doctors declared noticing increased opportunities in their own sector of employment by a larger margin (72.2%, 95% CI 66.1% to 77.7%), and private sector doctors confirmed the reduction of opportunities in private doctor practices

**Table 2** Physicians' perceptions of employment opportunities in public and private facilities, by sector of employment and state

| Employment opportunities | | Total (n=1183) | | São Paulo (n=632) 53.4% (50.6% to 56.3%) | | Maranhão (n=551) 46.6% (43.7% to 49.4%) | | P value* |
|---|---|---|---|---|---|---|---|---|
| | | % | 95% CI | % | 95% CI | % | 95% CI | |
| Public system (SUS) | Increased | 54.9 | 52.0 to 57.7 | 46.0 | 42.2 to 49.0 | 65.0 | 60.9 to 68.9 | <0.001 |
| | Reduced | 15.6 | 13.6 to 17.7 | 18.4 | 15.5 to 21.5 | 12.3 | 9.8 to 15.3 | |
| | No changes | 22.5 | 20.3 to 25.0 | 24.4 | 21.1 to 27.8 | 20.5 | 17.3 to 24.0 | |
| | Unknown | 7.0 | 5.7 to 8.6 | 11.2 | 9.0 to 13.9 | 2.2 | 1.2 to 3.7 | |
| Private doctor practices | Increased | 15.7 | 13.7 to 17.9 | 12.8 | 10.4 to 15.6 | 19.1 | 15.9 to 22.5 | 0.013 |
| | Reduced | 57.2 | 54.4 to 60.0 | 60.9 | 57.1 to 64.7 | 53.0 | 48.8 to 57.1 | |
| | No changes | 20.3 | 18.1 to 22.6 | 19.8 | 16.8 to 23.0 | 20.9 | 17.6 to 24.4 | |
| | Unknown | 6.8 | 5.4 to 8.3 | 6.5 | 4.8 to 8.3 | 7.0 | 5.2 to 9.4 | |
| Large private hospitals | Increased | 46.7 | 43.9 to 49.6 | 42.4 | 38.6 to 46.3 | 51.7 | 47.6 to 55.9 | 0.008 |
| | Reduced | 24.4 | 22.0 to 26.9 | 27.1 | 23.7 to 30.6 | 21.4 | 18.1 to 25.0 | |
| | No changes | 19.8 | 17.6 to 22.1 | 20.3 | 17.3 to 23.5 | 19.3 | 16.1 to 22.7 | |
| | Unknown | 9.1 | 7.5 to 10.8 | 10.2 | 8.1 to 12.8 | 7.6 | 5.6 to 10.1 | |
| Walk-in clinics | Increased | 25.2 | 22.9 to 27.8 | 24.7 | 21.4 to 28.2 | 26.0 | 22.4 to 29.7 | <0.001 |
| | Reduced | 19.9 | 17.7 to 22.2 | 16.9 | 14.2 to 20.0 | 23.2 | 19.9 to 26.6 | |
| | No changes | 24.0 | 21.6 to 26.5 | 20.4 | 17.4 to 23.7 | 28.1 | 24.5 to 32.0 | |
| | Unknown | 30.9 | 28.3 to 33.5 | 38.0 | 34.3 to 41.4 | 22.7 | 19.3 to 26.3 | |

Source: USP-UFMA-QMUL (2022).
*P value based on $X^2$ test.
SUS, Sistema Único de Saúde; USP-UFMA-QMUL, University of São Paulo-Federal University of Maranhão-Queen Mary University of London.

by 62.7% (95% CI 56.7% to 68.4%). For those doctors working concomitantly in public and private facilities—the dual practitioners—job opportunities increased in the public sector and in large private hospitals (56.4% and 46.3%, respectively) but reduced in smaller doctor practices (62.5%, 95% CI 58.8% to 66.1%) and stayed unchanged in walk-in clinics (see online supplemental table 2).

Regarding the availability of new doctors to take up vacancies in specific sectors, there was a slight perception of increased availability among public health physicians (43.1%, 95% CI 36.4% to 49.1%), although this was predominantly driven by the positive perceptions of Maranhão doctors (51.4%, 95% CI 43.2% to 59.5%)—among São Paulo doctors, this perception was, in fact, neutral if not negative (see table 3 below). Perceptions of increased availability of doctors were also recorded among dual practitioners in Maranhão (39.1%, 95% CI 37.8% to 48.2%). For private health physicians, perceptions of positive changes were only significant for Maranhão doctors (43.5%, 95% CI 31.7% to 55.9%).

Such reported changes in availability of vacancies and doctors, however, were not reflected in the perception of changes in prices; the largest group of doctors across all sectors (39.1%, 95% CI 32.4% to 47.6%) declared that remuneration for a 12-hour A&E shift stayed broadly

unchanged during the pandemic (see online supplemental table 3).

When disaggregating responses by age groups, younger doctors (aged 24–34 years) were the ones declaring increased job opportunities, particularly in the public sector (64%, 95% CI 58.1% to 68.1%) (see figure 1 below).

In private walk-in clinics, however, it was only the older doctors (>60 years) who reported significantly improved work opportunities (47.5%, 95% CI 39.9% to 55.1%), with all the other age groups reporting either decreased or unchanged employment opportunities (figure 1).

Doctors working directly with COVID-19 cases generally reported increased opportunities, particularly in public hospitals (65%, 95% CI 62.3% to 68.4%) and in large private ones (55%, 95% CI 51.8% to 59.1%) (see figure 2). Walk-in clinics were the only exceptions, as in such facilities specialising in working with patients with COVID-19 did not appear to significantly increase job opportunities (35.9%, 95% CI 32.1% to 40.0%).

## DISCUSSION

In our survey of labour market perceptions during COVID-19 among physicians in Brazil, we found that most doctors recounted increased job opportunities in

**Table 3** Perceptions of availability of doctors in each sector, by type of current public, private and dual employment

| Perception of availability of doctors | Total (n=1183) | | São Paulo (n=632) 53.4% (50.6% to 56.3%) | | Maranhão (n=551) 46.6% (43.7% to 49.4%) | | |
|---|---|---|---|---|---|---|---|
| | % | 95% CI | % | 95% CI | % | 95% CI | P value* |
| **Public sector (SUS)** | | | | | | | |
| Increased | 43.1 | 36.4 to 49.1 | 28.9 | 20.3 to 38.8 | 51.4 | 43.2 to 59.5 | 0.006 |
| Reduced | 32.3 | 25.3 to 37.2 | 37.8 | 28.3 to 48.1 | 26.8 | 20.0 to 34.5 | |
| No changes | 5.1 | 2.5 to 8.1 | 28.9 | 20.3 to 38.8 | 16.9 | 11.4 to 23.7 | |
| Unknown | 19.5 | 16.6 to 27.2 | 4.4 | 1.5 to 10.2 | 4.9 | 2.2 to 9.4 | |
| **Private** | | | | | | | |
| Increased | 38.7 | 31.8 to 43.5 | 35.7 | 29.3 to 42.5 | 43.5 | 31.7 to 55.9 | 0.475 |
| Reduced | 22.3 | 18.5 to 28.8 | 25.1 | 19.5 to 31.5 | 17.7 | 9.8 to 28.6 | |
| No changes | 7.9 | 5.8 to 12.7 | 29.6 | 23.6 to 36.3 | 32.3 | 21.6 to 44.5 | |
| Unknown | 31.1 | 24.9 to 36.0 | 9.5 | 6.0 to 14.2 | 6.5 | 2.2 to 14.6 | |
| **Dual practice** | | | | | | | |
| Increased | 39.1 | 36 to 43.3 | 36.2 | 31.2 to 41.3 | 43.0 | 37.8 to 48.2 | 0.057 |
| Reduced | 30.3 | 27.1 to 33.9 | 29.2 | 24.5 to 34.1 | 31.7 | 27 to 36.7 | |
| No changes | 5.9 | 4.6 to 8.2 | 26.8 | 22.3 to 31.7 | 20.7 | 16.7 to 25.2 | |
| Unknown | 24.7 | 20.7 to 27.0 | 7.8 | 5.4 to 11.1 | 4.6 | 2.8 to 7.2 | |
| **Total** | | | | | | | |
| Increased | 39.7 | 37 to 42.5 | 35.0 | 31.3 to 38.7 | 45.4 | 41.1 to 49.4 | ≥0.001 |
| Reduced | 29.0 | 26.5 to 31.6 | 29.1 | 25.7 to 32.7 | 28.7 | 25.2 to 32.7 | |
| No changes | 24.8 | 22.4 to 27.3 | 28.0 | 24.6 to 31.6 | 21.3 | 17.8 to 24.6 | |
| Unknown | 6.5 | 5.2 to 8.0 | 7.9 | 6.0 to 10.2 | 4.6 | 3.3 to 6.9 | |

Source: USP-UFMA-QMUL (2022).
*P value based on $X^2$ test.
SUS, Sistema Único de Saúde; USP-UFMA-QMUL, University of São Paulo-Federal University of Maranhão-Queen Mary University of London.

the public sector, particularly in Maranhão state. For the private sector, perceptions were mixed, as increased opportunities were reported in large private hospitals but not in smaller practices or walk-in clinics. In regard to the availability of doctors, our survey recorded perceptions of small increases in Maranhão, particularly in the public sector. Remuneration of A&E shifts stayed broadly unchanged. Younger doctors were the ones declaring more job opportunities in public facilities. Older ones reported opportunities in walk-in clinics, particularly in Maranhão. Those doctors working directly with patients with COVID-19 saw increases in SUS and large private hospitals, but not elsewhere.

We acknowledge that different doctors across the world experienced COVID-19 in a different way—from the frontline intensive care and infectious diseases specialists who found themselves in the eye of the storm, to primary care specialists who transitioned to remote working and telemedicine, to surgeons who simply saw their non-essential procedures cancelled. However, our survey of physicians' perceptions in Brazil during COVID-19 suggests that job opportunities actually increased in the public sector and in large private hospitals. This is contrast to what was observed for US hospitals during the pandemic.[20]

Our interpretation is that, in countries like Brazil with publicly funded health systems, resources (and jobs) were proactively redirected toward COVID-19 cases. This would explain why SUS and large private hospitals with inpatient care capacity in Brazil appear to have experienced additional vacancies to meet the increased demand for COVID-19 care. Conversely, smaller private health facilities with mostly outpatient capacity may have temporarily suspended some of their operations during the pandemic, in connection with the slowdown in demand for elective procedures. This would be consistent with what was observed in public health systems in European countries.[22]

Labour markets in São Paulo and Maranhão appear to have been affected in different ways by the pandemic, with the latter state seeing an increase of services and employment opportunities, and São Paulo doctors witnessing reduced activities and opportunities. This may be due

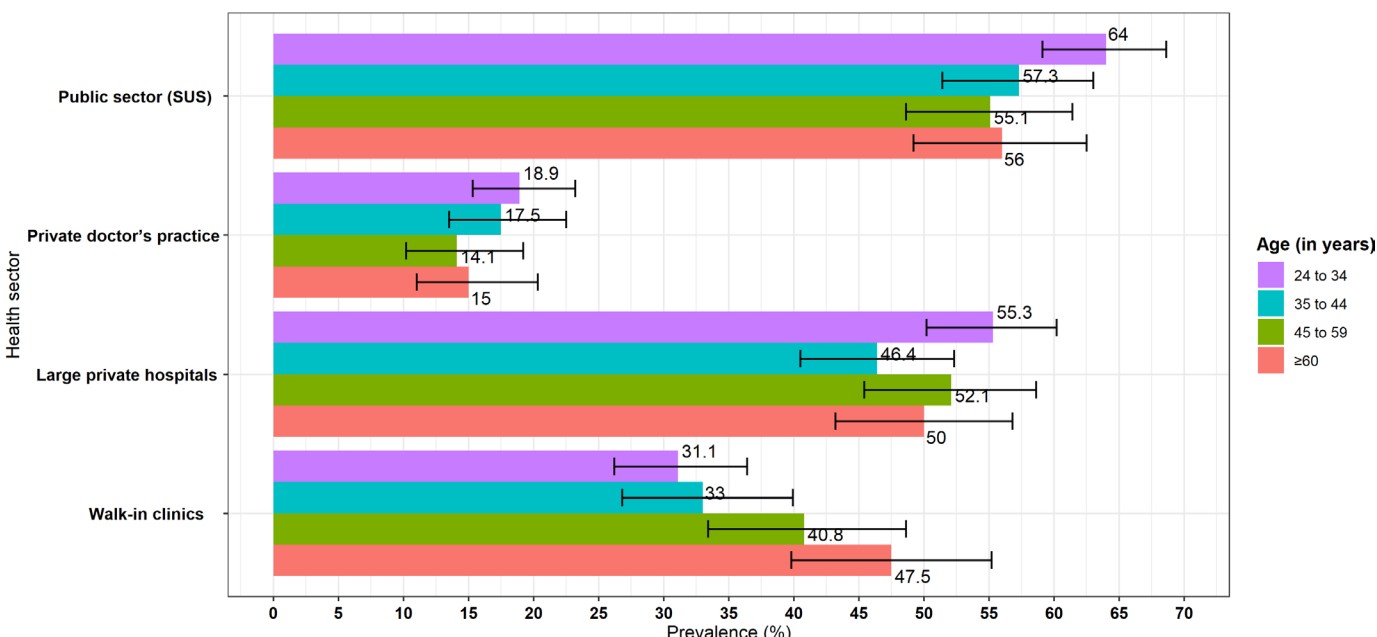

**Figure 1** Proportion of doctors reporting increased job opportunities, by age group and type of facility. Source: USP-UFMA-QMUL (2022). SUS, Sistema Único de Saúde; USP-UFMA-QMUL, University of São Paulo-Federal University of Maranhão-Queen Mary University of London.

to the comparatively greater weight of public health services in Maranhão's health system. In São Paulo, the private healthcare sector is very developed,[44] with an estimated 86% of doctors engaging with it either exclusively or as dual practitioners.[45] We conclude that the São Paulo health labour market effects experienced during COVID-19 were similar to the market in the USA, while Maranhão displayed features more like the European, Canadian or Australian markets. Such effects were probably exacerbated by the scarcity of doctors in Maranhão,[46]

who inevitably ended up taking more responsibilities (and risks) in the fight against COVID-19.[38]

Younger doctors reported increasing job opportunities across the board, particularly in the public sector and in Maranhão. We believe this reflects the decisions taken in Brazil—like in other countries—to deploy younger (and therefore less at risk) cadres to staff COVID-19 services and shelter more senior ones.[38] This would also be consistent with our findings on the increased opportunities in COVID-19 wards for younger doctors. We interpret the

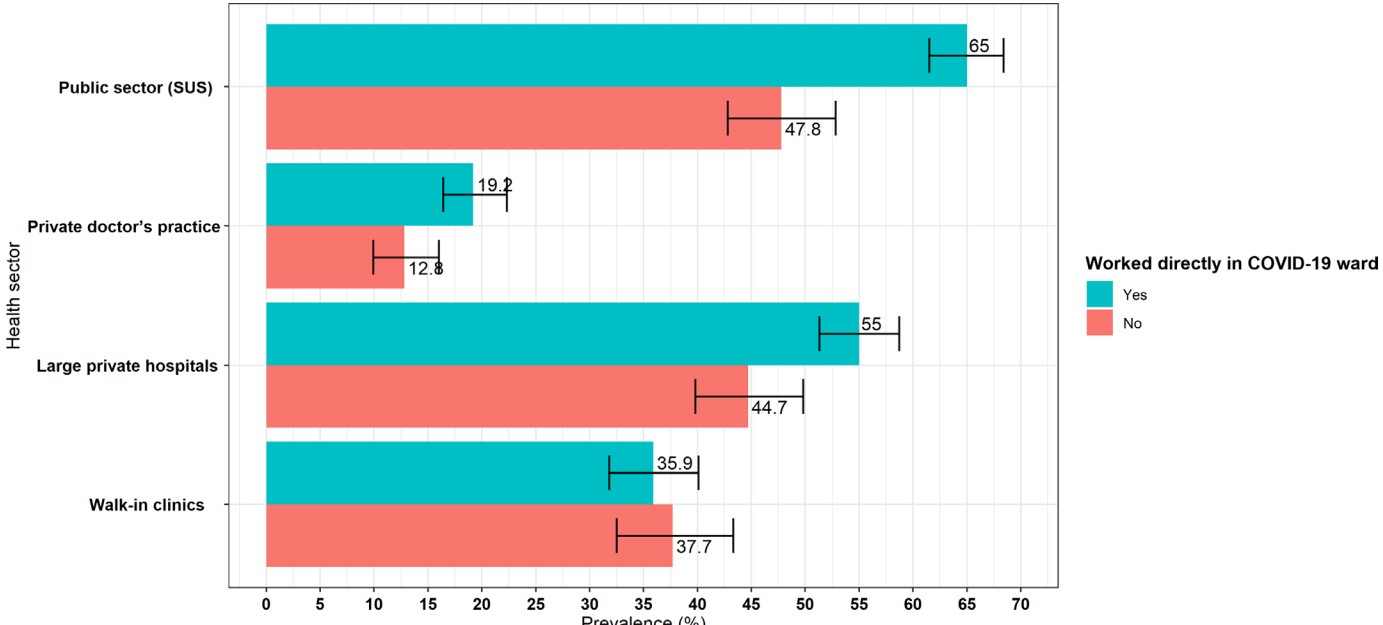

**Figure 2** Proportion of doctors declaring increased opportunities, by specialisation in COVID-19 cases and type of facility. Source: USP-UFMA-QMUL (2022). SUS, Sistema Único de Saúde; USP-UFMA-QMUL, University of São Paulo-Federal University of Maranhão-Queen Mary University of London.

reported increase of opportunities for older doctors in walk-in clinics as an indication that more lucrative parts of the private market would still be primarily accessible to more senior, established physicians, with fewer opportunities for younger doctors.[15]

Reports of no changes in remuneration for A&E shifts appear to be at odds with standard economic theory that would predict an increase of prices in the presence of increased demand and stable supply of doctors.[47] On the one hand, we acknowledge that remuneration for 12-hour A&E shifts may not be a suitable bellwether for changes in equilibrium prices for medical services (see the Limitations section below). On the other hand, it may be possible that, in the short run, labour prices for medical services may prove inelastic,[48] particularly during a pandemic emergency.

Our findings have broader relevance for other countries and future epidemics. We showed that health labour markets do not necessarily shrink during outbreaks, and the impacts will depend on the balance of public and private services within national health systems. Public health systems (and physicians) around the world were a key pillar of policy response to the pandemic, opening new services, performing additional functions and driving the clinical fight.[49] This inevitably poses questions on what the role of markets and the private sector should be during a public health emergency,[50] calling for a reconfiguration of the complementarity of public and private functions, with a view to boosting pandemic preparedness in LMICs.[51]

### Limitations

Our findings need to be interpreted in light of a number of limitations. First, we used physicians' perceptions of changes in vacancies and prices to gain an insight into the demand and supply of doctors during the pandemic in Brazil. Although workers' perceptions have been used before in the literature to explore labour market dynamics,[41] we acknowledge that an examination of employment data would be needed to validate our findings. We also asked our sampled physicians to report on changes that happened in the past 2 years, which could have been affected by recall bias.[52] Furthermore, a substantial proportion of doctors in Brazil concomitantly work in different hospitals and services, and the public/private nature of the entity running the services is not always known to them[42]; as a result, some of the 'public physicians' may have been erroneously classified, as they may be in reality working for 'private entities' and vice versa. We acknowledge this may have to some extent distorted our findings on the reaction of private and public labour markets during the pandemic.

Second, our proxies for demand, supply and price levels may have left too much room for interpretation, as some of our doctors struggled to distinguish between 'availability of job opportunities' and 'availability of doctors to fill vacancies'. Regarding the latter, we recognise that supply and recruitment difficulties would be better

known by hiring administrators and executives responsible for running hospitals, rather than by physicians. Our question on changes in remuneration for A&E shifts was driven by the need to identify a price indicator for medical services that could be known to all the doctors surveyed.[53] However, we realise that not all doctors in Brazil carry out A&E shifts or necessarily have knowledge of changes in this price. Such an anomalous finding on price stability may also be driven by erroneous measurement of supply, or by the complexity of adjusting prices for services in the public sector.

Finally, we recognise that Maranhão and São Paulo states present very particular configurations of labour market characteristics, organisation of health services, policies and health workforce.[32 37 42] Therefore, our findings may not be entirely generalisable to other LMICs.

### CONCLUSION

Limited evidence exists on health labour markets' impacts and adaptations during COVID-19, with some literature suggesting a reduction of services. We conducted a secondary analysis of survey data on physicians' perceptions around changing employment opportunities in one rich and one less developed state in Brazil in 2021, with the objective of gaining insights into health labour markets during epidemics in LMICs.

Most of our sampled doctors noticed increased job opportunities in the public sector, particularly in Maranhão state. For the private sector, perceptions were mixed, as increased opportunities were reported in large private hospitals but not in smaller clinics. Younger doctors perceived an increase of vacancies in public and in large private hospitals, while older ones reported opportunities in walk-in clinics, particularly in Maranhão. Those doctors working directly with patients with COVID-19 saw increases in public and large private hospitals, but not elsewhere.

Our findings suggest that health labour markets may not necessarily shrink during epidemics, and that the impacts will depend on the balance of public and private services in national health systems. The complementary roles of health markets and of publicly and privately funded systems during a health emergency should be re-examined, with the objective of improving pandemic preparedness, particularly for LMICs.

**Contributors** GR contributed to conceptualisation, participated in data analysis and drafted the manuscript. BLCALdO contributed to conceptualisation, participated in data analysis and reviewed the manuscript. AJFC participated in data collection, participated in data analysis and reviewed the manuscript. MCS contributed to conceptualisation and reviewed the manuscript. All the authors read and approved the final manuscript. GR is the guarantor of the overall content.

**Funding** This study, on whose findings this paper is based, received support from the Confap-MRC call for Health Systems Research Networks. Specifically, BLCALdO was funded by the Fundação de Pesquisa do Estado Maranhão (FAPEMA, Processo COOPI-00709/18) and Coordenação de Aperfeiçoamento de Pessoal de Nível Superior-Brasil (CAPES) (Finance Code 001), to the Graduate Program in Public Health. GR received an award from the Newton Fund, Medical Research Council (UK) (grant reference MR/R022747/1). MCS and AJFC received an award

from the Fundação de Amparo à Pesquisa do Estado de São Paulo (FAPESP-Brazil) (2017/50356-7).

**Disclaimer** The funders had no role in the survey's design, implementation or analysis.

**Competing interests** None declared.

**Patient and public involvement statement** Patients and/or the public were involved in the design, or conduct, or reporting, or dissemination plans of this research. Refer to the Methods section for further details.

**Ethics approval** This study involves human participants. The original survey received approval from the Research Ethics Committees of the Federal University of Maranhão, Brazil (CEP UFMA 3.051.875) and from the Faculty of Medicine of the University of São Paulo, Brazil (CEP FMUSP 3.136.269). The doctors surveyed were informed about the objectives of the study, guaranteed the anonymity of the information provided and their consent to participate obtained.

**Provenance and peer review** Not commissioned; externally peer reviewed.

**Data availability statement** Data are available upon reasonable request.

**ORCID iDs**
Giuliano Russo http://orcid.org/0000-0002-2716-369X
Bruno Luciano Carneiro Alves Luciano de Oliveira http://orcid.org/0000-0001-8053-7972
Mário C Scheffer http://orcid.org/0000-0001-8931-6471

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
