## [Reviewer comments · BMJ Open]

ARTICLE DETAILS

TITLE (PROVISIONAL)	What happened to health labour markets during COVID-19? Insights from the analysis of cross-sectional survey data on the perceptions of medical doctors in Brazil
AUTHORS	Russo, Giuliano; de Oliveira, Bruno Luciano Carneiro Alves; Cassenote, Alex; Scheffer, Mário

VERSION 1 – REVIEW

REVIEWER	Petre, Ionut Laurentiu Bucharest University of Economic Studies
REVIEW RETURNED	31-May-2023

GENERAL COMMENTS	Dear authors, I appreciate your work as interesting and topical. Please find below my main suggestions: Please provide statistical data on the determined sample, what is the accepted error? I recommend expanding empirical research and including survey results in this research, perhaps using some statistical software. This extension of empirical research can better support the discussions and conclusions presented. I appreciate the paper as well structured both from the point of view of the format and from the scientific point of view, I strongly recommend the detailed presentation of the empirical research and the implementation of the suggested changes in order to continue the path to publication.
--

REVIEWER	Rodriguez-Crespo, Ernesto Universidad Autónoma de Madrid
REVIEW RETURNED	04-Jun-2023

GENERAL COMMENTS	This paper examines an interesting research question, which is the relationship between COVID-19 and medical labor market in Brazil. I find important to shed more light on the following concerns: - It is necessary to acknowledge that COVID-19 impacts differed by countries in terms of economic development. It would also be important to clarify a short glimpse on stringency measures implemented in Brazil during COVID-19 and, more importantly, how these measures could potentially affect medical labor markets.- Apart from the survey content, which are the reasons to choose Maranhão and São Paulo against other Brazilian territories? You mention this idea in the manuscript but it shall be interesting for readers to include more descriptive information using figures, maps, tables,...
---

	- In relation to data, I would like to see a table summarizing the main descriptive statistics for each variable and disaggregated by territory. - Lastly, I believe the analysis could enrich by including a test of means difference between territories.
--	---

REVIEWER	Bhandari, Neeraj University of Nevada Las Vegas, Healthcare Administration and Policy
REVIEW RETURNED	09-Jun-2023

GENERAL COMMENTS	Manuscript review for BMJ Open Comments to authors Manuscript Number: BMJOpen-2023-075458 “What happened to health labour markets during COVID-19? Insights from a survey of medical doctors in Brazil” June 9, 2023 This study provides some intriguing survey data on perceptions of Brazilian doctors regarding demand for their services during the COVID-19 pandemic. The study is well written and uses simple but rudimentary descriptive methods to present its findings. It uses proxies for capturing perception of doctors, which limits its utility in two ways: perceptions can depart from reality and the specific proxies chosen here may contain some measurement error in and of themselves, especially the supply proxy items. That said the study is a good addition to existing body of literature on pandemic impacts. However, I would caution authors against reaching more sweeping interpretation than their data and methods allow. I am not sure the study results permit a clear inference about differential impacts of the pandemic on labor markets in private vs. the public sector, absent more information on the impacts on diverse settings within the public sector (e.g., primary care, urgent care elective care and inpatient settings). I discuss these (and other more minor) issues in more detail below:  1. It is not clear that the text of the question used to proxy physician supply is cleanly worded (“availability of physicians for recruiting”; Table 1 row 5) or directed at the right kind of subjects to elicit accurate information. Physicians are likely to have a good idea of demand for their own work but supply and recruitment difficulties would be better known by hiring administrators and executives responsible for running hospitals and clinics. Authors briefly allude to this issue in limitation but it needs to be emphasized more directly in interpreting their results. The anomalous finding of price stability given the supply-demand mismatch may be explained by an erroneous measurement of supply. 2. The text of the survey question for A&E remuneration differs between Table 1 (row 5, column 2) and the Datafolha instrument provided in the supplementary appendix (Q 34). The latter question does not refer to A&E shift but to specialty wages. Please address the discrepancy. 3. Please label supplementary tables using a different convention e.g., S2 or A2 to distinguish from tables in the main manuscript.
--

	4. The denotation for confidence interval is “CI” in the text and “IC” in the tables. Please make it consistent. 5. Were any formal statistical comparisons (t-test between group means) between responses for the two states made? I see it alluded in methods section but don’t see it discussed in results or described in tables. 6. I understand that responses to survey items included options to choose “unknown” when the respondent was unsure as to the main response options. However, I do not see this option listed in the descriptive data tables. Was this because no respondent picked that option? That would be unusual, especially for supply proxies, where the text is somewhat ambiguous. Relatedly, some of the cells in Table 3 do not add to 100%. For example, response options for overall sample regarding the perception of availability of doctors do not add up to 100% (increased 44.8, decreased 27.7, no change 19.2). Also, total number of respondents is listed as 1181 in some tables, whereas the Sao Paulo (632) and Maranhao respondents (551) total up to 1183. Please clarify these anomalies. 7. From the Datafolha instrument, it is evident that respondents were also asked to quantify the intensity of their initial responses to demand and supply proxy items on a 10-point scale (Q31a, 31b, 33a, 33b). Why was this information not utilized to further evaluate and contextualize their initial responses? 8. Authors interpret their findings as providing evidence that pandemic had differential effects on public sector and private sectors labor markets. I am not so sure. It may also be that pandemic had major impacts on specific acute care clinical settings like hospitals (where demand accelerated due to exploding COVID-19 related morbidity/mortality) regardless of whether these were public hospitals under SUS or private hospitals. Moreover, as authors themselves state, many SUS hospitals are run by private non-profit entities that hire contractual workers. Did SUS facilities that primarily provide elective or primary care buck the trend of reduced work opportunities? If so, that would support the theory of differential impacts but I am not sure whether SUS-related response data for demand proxy items was disaggregated by facility setting type 9. Some statements in the discussion are ambiguously worded and inconsistent with results: Demand and supply of doctors in Sao Paolo and Maranhao appear to have experienced opposite pandemic effects....., Page 10, Lines 59-60). Please review and reword.
--	---

VERSION 1 – AUTHOR RESPONSE

Reviewer: 1

Dr. Ionut Laurentiu Petre, Bucharest University of Economic Studies

Comments to the Author:

Dear authors,

I appreciate your work as interesting and topical. Please find below my main suggestions:

Please provide statistical data on the determined sample, what is the accepted error?

OUR RESPONSE: We have now introduced a table with the physicians sample in the statistical

annex (Online resources 2, Table S1). The sampling was explained in the Methods (Study settings and data collection, 2nd paragraph). For the tables in the main text, we only presented statistics with a p-value <0.05. We have now inserted the descriptive statistics and p-values from the statistical online resources to the main article file (see Table 2 in Results section, pag.8).

I recommend expanding empirical research and including survey results in this research, perhaps using some statistical software. This extension of empirical research can better support the discussions and conclusions presented.

OUR RESPONSE: We have now strengthened the statistical analysis by including p-values, confidence intervals for the association of each variable, and weight of non-respondents (see Tables 2 and 3 in main text). We have also strengthened the discussion of the internal and external validity of our results (see the bullet points in pag.2, and Discussion, final three paragraphs in pag.12).

I appreciate the paper as well structured both from the point of view of the format and from the scientific point of view, I strongly recommend the detailed presentation of the empirical research and the implementation of the suggested changes in order to continue the path to publication.

OUR RESPONSE: We have now presented more details of the empirical research carried out previously on this very sample of physicians (see Introduction, last but one paragraph in pag.4). We have also strengthened the statistical analysis for this paper (see the Variables and Data analysis section final two paragraphs (pag 7 after Table 1), as well as the new Table 2 and 3.

Reviewer: 2

Comments to the Author:

This paper examines an interesting research question, which is the relationship between COVID-19 and medical labor market in Brazil. I find important to shed more light on the following concerns:

- It is necessary to acknowledge that COVID-19 impacts differed by countries in terms of economic development. It would also be important to clarify a short glimpse on stringency measures implemented in Brazil during COVID-19 and, more importantly, how these measures could potentially affect medical labor markets.

OUR RESPONSE: We have now provided such background and references on the economy during COVID-19 in Brazil, on the introduction of lockdown measures, and their presumed impact on labour markets (Introduction, pag. 4, paragraphs 2 and 3).

- Apart from the survey content, which are the reasons to choose Maranhão and São Paulo against other Brazilian territories? You mention this idea in the manuscript but it shall be interesting for readers to include more descriptive information using figures, maps, tables,...

OUR RESPONSE: We selected Maranhão and São Paulo states as a way to contrast the impact of COVID-19 in one rich and developed state, with the impact on a much poorer state. We have explained this in the Study Settings section, pag.5, 1st paragraph). We also make references to our other publications from this survey, where the differences between MA and SP are analysed with more depths (see below).

Andrietta, L.S., Levi, M.L., Scheffer, M.C., Alves, M.T.S.S. de B. e, Oliveira, B.L.C.A. de, Russo, G., 2020. The differential impact of economic recessions on health systems in middle-income settings: a comparative case study of unequal states in Brazil. *BMJ Global Health* 5, e002122.

<https://doi.org/10.1136/bmjgh-2019-002122>

de Oliveira, B.L.C.A., Andrietta, L.S., Reis, R.S., de Carvalho, R.H. de S.B.F., de Britto e Alves, M.T.S.S., Scheffer, M.C., Russo, G., 2022. The Impact of the COVID-19 Pandemic on Physicians' Working Hours and Earnings in São Paulo and Maranhão States, Brazil. *International Journal of Environmental Research and Public Health* 19, 10085. <https://doi.org/10.3390/ijerph191610085>

- In relation to data, I would like to see a table summarizing the main descriptive statistics for each variable and disaggregated by territory.

OUR RESPONSE: As suggested, we have now included in the main manuscript the descriptive statistics for the variables, disaggregated by the two states (see Tables 2 and 3). We have also added a description at the beginning of the Results, a geographical map depicting our sample in the two states in Online Resources 2 (see Fig.S1), and added a table in the statistical annex summarising the sample's characteristics (Table S1 in Online resources 2).

Lastly, I believe the analysis could enrich by including a test of means difference between territories. OUR RESPONSE: As our analysis for this paper only considers qualitative variables (and therefore means difference cannot be tested), following Revi.2's suggestion we have now applied the statistics from the chi-square test as a complement to the confidence interval analyses (see Table 2 and 3).

Reviewer: 3

Comments to the Author:

This study provides some intriguing survey data on perceptions of Brazilian doctors regarding demand for their services during the COVID-19 pandemic. The study is well written and uses simple but rudimentary descriptive methods to present its findings. It uses proxies for capturing perception of doctors, which limits its utility in two ways: perceptions can depart from reality and the specific proxies chosen here may contain some measurement error in and of themselves, especially the supply proxy items. That said the study is a good addition to existing body of literature on pandemic impacts.

However, I would caution authors against reaching more sweeping interpretation than their data and methods allow. I am not sure the study results permit a clear inference about differential impacts of the pandemic on labor markets in private vs. the public sector, absent more information on the impacts on diverse settings within the public sector (e.g., primary care, urgent care elective care and inpatient settings). I discuss these (and other more minor) issues in more detail below:

1. It is not clear that the text of the question used to proxy physician supply is cleanly worded ("availability of physicians for recruiting"; Table 1 row 5) or directed at the right kind of subjects to elicit accurate information. Physicians are likely to have a good idea of demand for their own work but supply and recruitment difficulties would be better known by hiring administrators and executives responsible for running hospitals and clinics. Authors briefly allude to this issue in limitation but it needs to be emphasized more directly in interpreting their results. The anomalous finding of price stability given the supply-demand mismatch may be explained by an erroneous measurement of supply.

Our response: This is a valid point, and we accept the translation of this question from Portuguese into English was not accurate. We have now changed it to 'Availability of new doctors to fill positions' (see table 1). We also acknowledge in the Discussion that such information on supply of doctors may not have been known to physicians (see Limitations, pag.12, 2nd paragraph), and included the answer 'Unknow' as answers in the tables. As for the anomalous finding on price stability, we offer a range of possible explanations, and have now added 'erroneous measurement of supply' to that list (same paragraph in Limitations at the end of Discussion).

2. The text of the survey question for A&E remuneration differs between Table 1 (row 5, column 2) and the Datafolha instrument provided in the supplementary appendix (Q 34). The latter question does not refer to A&E shift but to specialty wages. Please address the discrepancy.

OUR RESPONSE: This has now been addressed. The discrepancy was due to poor translation of the Datafolha questionnaire into English (see Q34 in Table 1 and Q34 in Online resources 1).

3. Please label supplementary tables using a different convention e.g., S2 or A2 to distinguish from tables in the main manuscript.

OUR RESPONSE: This has now been addressed, and the supplementary tables from the online resources 2 have now been called S1, S2 and S3.

4. The denotation for confidence interval is "CI" in the text and "IC" in the tables. Please make it consistent.

OUR RESPONSE: This has now been addressed in the new Table 2 and 3.

5. Were any formal statistical comparisons (t-test between group means) between responses for the two states made? I see it alluded in methods section but don't see it discussed in results or described in tables.

Our response: As in this manuscript we are only considering qualitative variables, t-test for means values cannot be calculated; however, we have now applied the statistics from the chi-square test as a complement to the confidence interval analyses (see the new Tables 2 and 3).

6. I understand that responses to survey items included options to choose "unknown" when the respondent was unsure as to the main response options. However, I do not see this option listed in

the descriptive data tables. Was this because no respondent picked that option? That would be unusual, especially for supply proxies, where the text is somewhat ambiguous. Relatedly, some of the cells in Table 3 do not add to 100%. For example, response options for overall sample regarding the perception of availability of doctors do not add up to 100% (increased 44.8, decreased 27.7, no change 19.2). Also, total number of respondents is listed as 1181 in some tables, whereas the Sao Paulo (632) and Maranhao respondents (551) total up to 1183. Please clarify these anomalies.

Our response: This is a very important point; in our previous analysis we had chosen (erroneously) to disregard the “unknown” category. We have now corrected this in the revised analysis, and inserted this category in the tables. As a result, also the key statistical results have been changed accordingly through the manuscript. We thank the reviewer for spotting this. Table 3 has also been corrected, as well as overall number of physicians included. Apologies for this oversight.

7. From the Datafolha instrument, it is evident that respondents were also asked to quantify the intensity of their initial responses to demand and supply proxy items on a 10-point scale (Q31a, 31b, 33a, 33b). Why was this information not utilized to further evaluate and contextualize their initial responses?

OUR RESPONSE: Response rate to those questions were too low to allow meaningful significant statistical analysis. We noticed physicians struggled to attribute a 0-10 score to the intensity of changes observed during the pandemic.

8. Authors interpret their findings as providing evidence that pandemic had differential effects on public sector and private sectors labor markets. I am not so sure. It may also be that pandemic had major impacts on specific acute care clinical settings like hospitals (where demand accelerated due to exploding COVID-19 related morbidity/mortality) regardless of whether these were public hospitals under SUS or private hospitals. Moreover, as authors themselves state, many SUS hospitals are run by private non-profit entities that hire contractual workers. Did SUS facilities that primarily provide elective or primary care buck the trend of reduced work opportunities? If so, that would support the theory of differential impacts but I am not sure whether SUS-related response data for demand proxy items was disaggregated by facility setting type

OUR RESPONSE: We accept this point about some public hospitals being run private entities (known as Social Health Organizations – OSS). However, as a substantial proportion of doctors concomitantly work on different hospitals and services, and the public/private nature of the entity running the services is not always know to them, it was not possible to collect such information. We have now acknowledged such limitation in the Discussion (Limitations, pag.12, 1st paragraph).

9. Some statements in the discussion are ambiguously worded and inconsistent with results: Demand and supply of doctors in Sao Paolo and Maranhao appear to have experienced opposite pandemic effects....., Page 10, Lines 59-60). Please review and reword.

OUR RESPONSE: This sentence has now been reworded (see the Discussion, pag.11, 2nd paragraph).

We would like to thank the three reviewers for spotting the inaccuracies in our earlier draft, and for offering solutions to fix them.

VERSION 2 – REVIEW

REVIEWER	Rodriguez-Crespo, Ernesto Universidad Autónoma de Madrid
REVIEW RETURNED	06-Jul-2023

GENERAL COMMENTS	My comments have been amended. I recommend the paper for publication.
---

REVIEWER	Bhandari, Neeraj University of Nevada Las Vegas, Healthcare Administration and Policy
-----------------	--

REVIEW RETURNED	21-Jul-2023
GENERAL COMMENTS	I thank the authors for addressing major concerns identified in the first review . I have no further comments